# Management and Outcomes of Tibial Eminence Fractures in the Pediatric Population: A Systematic Review

**DOI:** 10.3390/children10081379

**Published:** 2023-08-13

**Authors:** Beatrice Limone, Francesco Zambianchi, Giorgio Cacciola, Stefano Seracchioli, Fabio Catani, Luigi Tarallo

**Affiliations:** 1Department of Traumatology, Orthopaedic and Occupational Medicine, CTO Hospital, University of Study of Turin, 10124 Turin, Italy; 2Department of Orthopaedic Surgery, Azienda Ospedaliero Universitaria di Modena, University of Modena and Reggio-Emilia, 41121 Modena, Italy

**Keywords:** tibial eminence fractures, children, surgery, arthroscopic, open reduction, systematic review

## Abstract

Background: Tibial eminence fractures (TEF) of Meyers–McKeever type II-III-IV usually require surgical management. No consensus in the literature has been achieved regarding the best treatment option. The aims of the present systematic review were (1) to analyze the current literature and describe the outcomes of surgical treatment for TEF; and (2) to compare the outcomes of different surgical options using arthroscopic reduction and internal fixation (ARIF) with sutures or screws and open reduction and internal fixation (ORIF). Methods: A search was carried out with Pubmed, Medline, and Cochrane. Key terms were used “tibial” AND “eminence” or “spine” or “intercondylar” AND “paediatric” or “children” AND “fracture” or “avulsion” AND “treatment”. Twelve articles met the inclusion criteria. Demographic data, clinical outcomes, and complication rates were evaluated for each study. Means/standard deviation and sum/percentage were used for continuous and categorical variables, respectively. Chi-square or t-student tests were applied. A *p*-value < 0.05 was considered statistically significant. Results: ORIF showed superior clinical outcomes (Tegner (*p* < 0.05) and Lysholm (*p* < 0.001) scores) relative to ARIF and a lower incidence of arthrofibrosis (*p* < 0.05) and implant removal (*p* < 0.01). The Tegner, IKDC, and Lysholm scores showed statistically significant superior results following arthroscopic sutures compared to arthroscopic screws (*p* < 0.001). The incidence of arthrofibrosis was higher after arthroscopic sutures (*p* < 0.05), the implant removal was higher after screw fixation (*p* < 0.001) Conclusions: Better clinical results with low complication rates were achieved with ORIF surgery rather than ARIF; arthroscopic suture fixation resulted in higher clinical results compared to arthroscopic screw fixation and reduced the incidence of postoperative complications.

## 1. Introduction

Tibial eminence fractures (TEFs) consist of bony avulsions of the anterior cruciate ligament (ACL) from its distal insertion. Due to the incomplete ossification of the tibial eminence in children, damage occurs to the bone rather than the ligament when traumatic force is applied [1]. The incidence is higher in the male gender and the peak age is between 13 and 14 years old [2].

These injuries commonly occur after a fall from a bicycle or during sports activities, resulting in a pivot-type rotation mechanism of injury. A combination of hyperextension and internal rotation leads to knee twisting and consequently intercondylar eminence avulsion [3]. Occasionally, TEFs can also occur because of direct trauma or hyperextension of the joint.

Patients usually present at the emergency department complaining about knee pain, severe swelling, and inability to bear weight. After clinical examination, diagnosis is performed using standard anteroposterior and lateral knee radiographs. As small fragments can be difficult to visualize, leading to misdiagnosis, a computed tomography (CT) scan is useful for characterizing bony fragments and fracture extension and for performing adequate preoperative planning in displaced fractures requiring operative treatment.

TEFs have been historically classified according to the Meyers and McKeever classification system, based on plain radiographs. Type I are non-displaced, or minimally displaced fractures (<3 mm), involving the anterior margin of the spine. Type II fractures present minimal superior displacement of the anterior bony fragment with an intact posterior cortical hinge. Type IIIA are completely displaced, but not rotated; Type IIIB present displacement and fragment rotation. Type IV, later described by Zaricznyj, are displaced and comminuted [4]. More recently, Green et al. introduced a new MRI-based classification system, providing specific, quantitative criteria for classifying fractures according to fragment displacement and tissue involvement, assisting clinicians with subsequent treatment decisions [5].

For nondisplaced or minimally displaced Meyers–McKeever type I fractures, conservative treatment is generally advised with immobilization in a cast or splint in extension for 6–10 weeks to reduce fracture gap and enhance healing. Type II fracture treatment has been controversial, with initial closed reduction that can be attempted. If reduction is not adequately obtained, or in the case of redisplacement, operative treatment is usually recommended. An inadequate reduction can occur for the interposition of either the medial meniscus anterior horn, the lateral meniscus, or the transverse ligament within the fracture site. When surgical management is advised, both open and arthroscopic approaches can be used. Open reduction and internal fixation (ORIF) is usually performed with screws and represents an effective and safe surgical option with few complications. Screw insertion can be either retrograde or anterograde, with a simple and reproducible surgical technique, allowing for early mobilization and ambulation. The advantages of arthroscopic reduction and internal fixation (ARIF) include reduced soft-tissue trauma, direct fragment visualization, and the potential for simultaneous treatment of associated meniscal tears, meniscal and intermeniscal ligament entrapment, interstitial tears of the ACL, and removal of loose bodies [6]. Although tibial eminence fractures have an excellent prognosis, complications may occur with either treatment. The most common complications include arthrofibrosis, clinical knee instability, and the need for surgical removal of fixation hardware [7]. To date, the best treatment option is controversial and no consensus in literature is achieved to support one fixation method over any other.

The aim of the present systematic review was three-fold: (1) to analyze current literature and describe the overall outcomes of children who underwent surgical treatment for Type II-III-IV TEF; (2) to compare the outcomes of treatment for ARIF and ORIF; and (3) to compare arthroscopic screw fixation with arthroscopic suture fixation.

## 2. Material and Methods

### 2.1. Search Criteria

The present review was performed in accordance with the Preferred Reporting Items for Systematic Reviews and Meta-Analyses (PRISMA) guidelines [8]. A comprehensive search was carried out with Pubmed, Medline, and Cochrane for randomized controlled trials, quasi-randomized controlled trials, and prospective/retrospective cohort studies evaluating the clinical outcomes of pediatric patients treated for TEF. Various combinations of search terms were used, such as “tibial” AND “eminence” or “spine” or “intercondylar” AND “paediatric” or “children” AND “fracture” or “avulsion” AND “treatment”. Only abstracts dealing with pediatric fracture treatment were evaluated. The reference lists of all identified studies were queried for additional eligible studies.

### 2.2. Inclusion and Exclusion Criteria

The inclusion criteria were (1) studies including a minimum of 5 patients with TEF, (2) a pediatric population (age < 16 years), (3) Meyers–McKeever type II-III-IV TEF with surgical management, (4) absence of associated major knee injuries, (5) ORIF or ARIF with screws or sutures, and (6) publication in the English language.

The exclusion criteria were (1) articles not meeting the inclusion criteria, (2) case reports, (3) articles describing surgical techniques, and (4) fracture fixation performed with absorbable pins.

### 2.3. Study Screening

Two independent reviewers carried out comprehensive research regarding the abstracts by applying the previously described criteria. If the title and abstract were deemed relevant by either reviewer, the article was reviewed in full text. After this stage, other studies were excluded if the full text did not meet the eligibility criteria (Figure 1). A total of 12 studies met eligibility requirements and were divided based on different surgical approaches. If both open and arthroscopic procedures were included, demographic, clinical, and outcome data were reported separately for different surgical techniques (Table 1).

### 2.4. Data Collection

During the review, the following information was collected for each study: title, first author, year of publication, follow-up, number of patients, demographic data (age, gender), injury etiology, type of surgical treatment, fracture’s Meyers–McKeever classification type, the number of days between trauma and surgery, clinical score evaluation (Tegner score, KT-1000 score, International Knee Documentation Committee (IKDC) score, Lysholm score), and complications after surgery.

The clinical evaluation was performed using the following tests. The Tegner scale is useful to assess daily activities and knee function after surgery [9]. It is a numerical scale from 0 to 10, and each number represents a different level of activity. The higher the number, the higher the activity level.

The IKDC score, ranging from 0 to 100, where 100 means no limitations in daily and sports activities, is another common PROM (several patients’ reported outcome measures) to evaluate postoperative knee function [10]. The KT-1000 arthrometer is a tool designed to quantify anterior tibial shifting relative to the femur after ACL (anterior cruciate ligament) repair. As the ACL is the primary restrain to tibial anterior shift, the test is appropriate to evaluate ACL repair, using it as a dichotomous variable with a threshold of 2 to 3 mm [11].

The Lysholm score is a valid patient-administered test that assigns a score to eight parameters (pain, instability, locking, swelling, limping, stair climbing, squatting, and the need for support) ranging from 0 to 100, with 100 representing no disability [12].

**Table 1 children-10-01379-t001:** Studies characteristics and demographic data of patients treated for tibial eminence fractures. * Studies reporting more than one surgical treatment; YoP = year of publication; SD = standard deviation; FU = follow-up; NA = not available.

First Author (YoP)	Patient Initially/Final	Type of Treatment	Age/SD or Range (Year)	FU/SD or Range (Months)	Male (%)	Type of Trauma, No of Patients	Complications
Zheng (2021) * [11]	10/10	Open	10.9/1.53	27.1/11.8	50%	NA	NA
	24/24	Arthroscopic screw	10.9/2	27.5/11.8	45.8%	NA	NA
Edmonds (2015) * [12]	29/29	Open	12.2/3	81.6/24	82.7%	NA	Arthrofibrosis
	28/28	Arthroscopic suture	12.4/2	81.6/24	64.3%	NA	Arthrofibrosis
Watts (2016) * [13]	13/13	Open	11.5/2.6	12.7/14.3	53.8%	NA	Arthrofibrosis
	8/8	Arthroscopic screw	12.9/2.7	13.9/10.5	na	NA	NA
	9/9	Arthroscopic suture	12.9/2.7	13.9/10.5	na	NA	NA
Xu (2016) [14]	21/21	Arthroscopic suture	15.3/13–17	43.4/40–47	71.4%	Sport (9), car accident (8), other (3)	NA
Zhao (2018) [15]	22/22	Arthroscopic suture	11.3/8–16	34.5/24–46	59%	NA	NA
Callanan (2019) * [16]	33/33	Arthroscopic suture	12.4/12–15	24/NA	66.7%	Sport (30), other (3)	Arthrofibrosis, implant removal, other
	35/35	Arthroscopic screw	12.2/3.3	48/NA	77.1%	Sport (28), other (7)	Arthrofibrosis, implant removal, other
Çağlar (2021) [17]	28/28	Arthroscopic suture	14.2/8–18	55.7/28.8–87.6	60.1%	Other (28)	Arthrofibrosis
Russu (2021) [18]	12/12	Arthroscopic suture	14.3/2.1	6/NA	33%	Sport (12)	NA
Honeycutt (2020) [19]	35/35	Open	11.2/3.3	48/NA	77%	Sport (28), other (7)	Arthrofibrosis, implant removal, other
Chalopin (2022) [20]	20/17	Arthroscopic suture	12/7.15	28/16–48	47%	NA	NA
Zhang (2020) [21]	21/21	Arthroscopic suture	12.7/2.1	24/22.6–34	66.7%	NA	NA
Quinlan (2021) [22]	97/66	Arthroscopic suture	10.7/4–17	69.6/12–142.8	50%	NA	NA

### 2.5. Level of Evidence and Studies Quality Assessment

The Levels of Evidence, established by the Oxford Centre for Evidence-based Medicine, were used to assign the correct level of evidence to each study [13]. Data were collected and the quality of the studies was assessed with methodological index for non-randomized studies (MINORS) [14]. The index evaluates twelve items, the last four of which are specific for comparative studies: the aim of the study, inclusion of consecutive patients, prospective collection of data, appropriateness of the endpoints, unbiased assessment of the endpoint, appropriateness follow-up length, percentage of loss to follow-up, prospective calculation of the sample size, comparable control group, contemporary control groups, baseline equivalence of groups, and the adequateness of the statistical analysis. The sum of the 0 to 2 score assigned to each parameter was used to evaluate the quality of the study: poor (0–8 or 0–12 for non-comparative and comparative studies, respectively), good (9–12 or 13–18 for non-comparative and comparative studies, respectively), and excellent (13–16 or 19–24 for non-comparative and comparative studies, respectively).

Eight studies were classified as level of evidence IV [15,16,17,18,19,20,21,22] and 4 were classified as level of evidence III [23,24,25,26].

The overall mean MINORS score reported was 14.8. The average quality was 11.3 for non-comparative studies and 19.6 for comparative studies (Table 2).

### 2.6. Statistical Analysis

Descriptive characteristics of the study sample were calculated as the mean and the standard deviation (SD) for continuous variables and as the absolute value and percentage frequencies for categorical variables. Descriptive analyses were conducted using Chi-square or *t*-student tests, as appropriate, for categorical and continuous variables, respectively. Statistical analyses were performed with Microsoft Excel (Microsoft Corporation, Redmond, Washington, DC, USA), where the α significance level was 0.05 with a 95% confidence level.

## 3. Results

### 3.1. Demographic and Clinical Data of All Patients

A total of 416 knees were included for assessment. Thirty-five (8.4%) were excluded due to the loss of follow-up data, leaving a total of 381 cases available for analysis. The mean age of the study group at the time of surgery was 12.1 years (SD 1.3). There were 222 males (61%) and 142 females (39%) at a mean follow-up of 45.7 months (SD 23.9) postoperatively (Table 1). Five studies [18,20,21,22,23] reported the etiology of the trauma (177 knees). Sports injuries were the most frequent injury mechanism (113 cases—63.8%), followed by car/motorcycle accidents in 20 cases (11.3%), and other causes in 44 cases (25.1%). Eleven studies [15,17,18,19,20,21,22,23,24,25,26] reported the classification of the lesion according to Meyers–McKeever [4] (297 knees). The most frequent TEF pattern was type III in 153 knees (51.5%), followed by type II in 127 (42.8%), and type IV in 17 cases (5.7%). On average, as reported by nine studies [15,16,17,18,22,23,24,25] the number of days between trauma and surgical treatment was 7.9 (SD 4.9). The mean postoperative Tegner score was reported in three studies [15,17,22] at 7.2 (SD 0.8), and the IKDC score was reported in five studies [15,19,22,24,25] with a mean of 88.5 (SD 5.4), while the mean postoperative Lysholm score was 92.4 (SD 4.9) [15,16,18,19,22,23,24,25]. The overall complication rate considering all surgical treatments was 17.6% (67 cases/381) and the most frequent postoperative complication was knee arthrofibrosis.

### 3.2. Demographic and Clinical Data of ORIF

Four out of the twelve studies assessed described open surgical treatment [15,16,17,23] and included a total of 57 knees. The mean follow-up was 44.7 months (SD 22.5) and the mean patient age at the time of surgery was 11.7 years (SD 0.6). There were 40 males (85%) and 17 females (15%). One study described the etiology of TEF [23] and the two main reported mechanisms of trauma were sport injuries (two cases) and other causes (three cases). Fracture classification was described in three of the twelve studies (28 knees), and the most frequent fracture pattern was type III (17 cases, 60.7%). There was an average of 6.3 days (SD 2.9) between traumatic events and surgery. The Tegner and IKDC scores were reported in one study [15], with a mean value of 7.8 (SD 0.9) and 92.1 (SD 3.6), respectively. The Lysholm score was reported in three articles [15,16,23] with a mean value of 96.7 (SD 3.2). Three articles reported treatment complications [16,17,23], with arthrofibrosis resulting as the most frequent (8.5%).

### 3.3. Demographic and Clinical Data of Arthroscopic Treatment

Eleven studies describing arthroscopic techniques for the treatment of TEF were detected [15,16,17,18,19,20,21,22,24,25,26]. A total of 359 knees underwent arthroscopic sutures or arthroscopic-guided screw fixation and 325 of them were evaluated at a mean of 44.7 months (SD 22.5) of follow-up. The mean patients’ age at the time of surgery was 12.2 (SD 1.5). A total of 193 TEF occurred in male subjects (59%) and 132 in females (41%). Four studies [18,20,21,22] described the injury mechanism, with sports injuries accounting for TEF in 79 cases. Ten articles [15,17,18,19,20,21,22,24,25,26] described fractures according to the Meyers–McKeever classification. The most frequent was type III (141 cases, 49.1%), followed by type II (129 cases, 44.9%) and type IV (17 cases, 6%). On average, 8.6 days (SD 5.9) passed between the traumatic event and surgery. The Tegner score was reported in three studies [15,18,22] with a mean value of 7.1 (SD 0.9). One author [18] reported the KT-1000 test result, with a mean of 3.4 (SD 2). The IKDC score results were reported by six studies [15,19,22,24,25,26], with a mean of 89.7 (SD 5.2), and the Lysholm score was described in seven studies [15,16,18,19,22,24,25] and resulted on average 91.4 (SD 4.8). Thirty-two cases of arthrofibrosis (24.2%) were observed [16,17,20,21] and twenty-six cases required hardware removal for intolerance (18.3%).

### 3.4. Demographic and Clinical Data of Arthroscopic Suture Treatment

Ten studies described arthroscopic suture surgical procedures [16,17,18,19,20,21,22,24,25,26]. A total of 291 cases were initially included; 34 were lost at follow-up and the mean follow-up was 46.9 months (SD 24.4). The mean age of the cohort was 12.4 years (SD 1.4). There were 144 males (58%) and 104 females (42%). Etiology was described in four articles [16,17,20,21]: 51 cases of sports injuries, 8 cases of car/motorcycle accidents, and 34 of other causes were reported. The Meyers–McKeever classification was defined in eight articles [15,18,19,20,21,22,23,24]. Type II was the most frequent (107 cases, 48.6%), followed by type III (90 cases, 40.9%) and type IV (17 cases, 7.7%). There was a mean of 9.2 days (SD 6.4) between traumatic events and surgical procedures. Two studies reported the Tegner score [18,22] and the mean value was 7.5 (SD 0.9). KT-1000 results were reported in one study [18]. The IKDC score was analyzed in five studies [19,22,24,25,26], while the Lysholm score was reported in six studies [16,18,19,22,24,25], with a mean value of 90.4 (SD 5.6) and 92.5 (SD 4.6), respectively. Three articles described treatment complications [16,20,21], with arthrofibrosis resulting as the most frequent (eight cases).

### 3.5. Demographic and Clinical Data of ARIF

Three articles reported the results of 67 knees who underwent ARIF with screws [15,17,20]. The mean follow-up was 36.6 months (SD 17.2) and the mean age of the cases included for assessment was 11.3 (SD 1.1). Thirty-eight males (64%) and twenty-one females (26%) were assessed. The injury etiology was reported in one study [20] and the most frequent cause of TEF was sports injuries (28 cases, 80%). Two studies classified fractures according to the Meyers–McKeever system [15,20], with 46 cases of type III (77.9%) and 9 type II fractures (15.5%). On average, 6.3 days (SD 0) passed between traumatic events and surgical treatment. The Tegner score was described in one study [15], with a mean value of 6.4 (SD 0.5). The IKDC and Lysholm scores were reported in one study [15], with a mean value of 86.1 (SD 5.8) for the former and 86.2 (SD 4.5) for the latter.

### 3.6. Comparison between Open and Arthroscopic Treatment

Subjects treated with ORIF were on average younger (*p* < 0.05) and underwent surgery earlier (*p* < 0.01) compared to those who underwent ARIF. Open treatment revealed, on average, superior Tegner (*p* < 0.05) and Lysholm (*p* < 0.001) scores relative to arthroscopic procedures and was followed by a lower incidence of complications, with a lower incidence of arthrofibrosis (*p* < 0.05) and implant removal (*p* < 0.01) (Table 3).

### 3.7. Comparison between Arthroscopic Suture and Arthroscopic-Guided Screw Fixation Treatment

Differences were detected between studies describing arthroscopic-guided sutures and screw TEF fixation in terms of follow-up length and patients’ age (*p* < 0.001). Differences were also reported in terms of the etiology of trauma (*p* < 0.001) and days between trauma and surgery (*p* = 0.01). The Tegner, IKDC, and Lysholm scores showed statistically significant superior results following arthroscopic sutures compared to arthroscopic fracture fixation with screws (*p* < 0.001). While the incidence of knee arthrofibrosis was higher in knees undergoing arthroscopic sutures (*p* < 0.05), the need for implant removal was higher in cases undergoing screw fixation (*p* < 0.001) (Table 4).

## 4. Discussion

The present systematic review aimed to analyze the current literature and describe the overall outcomes of a pediatric population undergoing surgical treatment for TEF by comparing the results of different surgical procedures.

The incidence of TEF is higher in a pediatric population aged 10–14 years, with a mean value of 12.4 years. As reported by Skak et al., while metaphyseal fractures are predominant in younger children, eminence fractures, ligament ruptures, and pysheal injuries are more common in teenagers [27]. As reported in previous studies [2,28], TEF had higher incidence in male subjects (222 cases, 61%) compared to females (142 cases, 39%).

Sport activities represented the most common cause of tibial eminence fractures; other injury mechanisms were car accidents and falls from bicycles and trampolines [29]. In total, 81 cases (60.5%) of TEF resulting from the present systematic review were sports-related injuries. This result is in line with the incidence reported in the literature. It can be hypothesized that the recent increase in sports participation at younger ages and early sports specializations may play a role in the increased incidence of such injuries and more severe fracture patterns [3,30].

Several patient-reported outcome measures (PROMS) have been used to evaluate clinical and functional improvement after surgical treatment of tibial eminence fracture. While a healthy population has an average Tegner score of 5.7 [31], the mean Tegner score in the studies included in this review was 7.2 (SD 0.8), meaning that patients were able to participate in almost all sports at a high level after surgery. Comparing Tegner scores in subjects undergoing ORIF and ARIF, a statistically significant result in favor of the open technique was observed (*p* < 0.05). However, the Tegner score was evaluated in one study describing open surgery results [15], while three studies regarding arthroscopic procedures took this score into account [15,18,22]. In addition, some authors reported no difference in terms of clinical outcomes measured with the Tegner score between ARIF and ORIF [16,32]. The IKDC score showed no statistically significant differences when open and arthroscopic treatments were compared. The outcomes reported in the present review are in line with the findings reported by Shimber et al. [32], who demonstrated no differences between ORIF and ARIF in terms of IKDC results. Conversely, the arthroscopic suture technique was shown to achieve a higher level of IKDC score compared to arthroscopic-guided screw fixation. However, in their systematic review, Osti et al. reported no outcome differences for different fixation methods [33].

The KT-1000 arthrometer test was applied in one of the studies included in the present review and the mean value reported following arthroscopic suture fixation was 3.4 mm [18]. It may be presumed that even following anatomical tibial eminence reduction, a mild ACL laxity may remain postoperatively. A comparison between different surgical approaches in terms of anterior tibial shift was not possible, as not enough studies reported results of the KT-1000 test.

To emphasize the evaluation of postoperative knee instability, the Lysholm score is a valid patient-administered test. The mean value of the Lysholm score reported by the studies included in the present systematic review showed good results (92.4 SD 4.9). ORIF demonstrated a higher Lysholm score compared to the ARIF technique. Indeed, as Jääskelä et al. showed, children who underwent open surgery for TEF usually have a quicker return to play/sports [34].

In the context of arthroscopic procedures, the suture technique provided better Lysholm scores compared to screw fixation. The arthroscopic suture fixation technique allows for the restoration of joint congruity and tibial eminence integrity, achieving a postoperative full range of motion and return to daily activities [35].

The most common complication following TEF treatment consists of knee stiffness (arthrofibrosis), often resulting in knee pain, decreased function, and inability to return to sports [36]. In total, 30 knees (17.5%) developed arthrofibrosis after surgery, with a higher frequency after arthroscopic procedures compared to ORIF (*p* < 0.05). Arthrofibrosis represents one of the major complications of knee arthroscopy, often requiring additional surgical treatment [37] and, according to the findings of the present systematic review, had a higher incidence following arthroscopic sutures compared to screw fixation procedures (*p* < 0.05). This comparison is limited by the fact that only one study dealing with screw fixation reported the rates of joint arthrofibrosis after surgery. To prevent knee stiffness, inflammation control and early knee motion are generally advised [38]. Postoperative knee pain and function impairment can also be due to hardware intolerance [39]. In the present systematic review, a total of 26 cases (15.2%) required a second surgical procedure to remove fixation hardware. Higher rates of hardware intolerance after arthroscopic procedures were reported for arthroscopic procedures compared to ORIF (*p* < 0.01), especially when screws were the chosen fixation device (*p* < 0.001) [40]. The use of screws may lead to anterior impingement and potential damage to the femoral intercondylar notch, determining pain and requiring hardware removal [41].

Incomplete literature and a low level of evidence in the studies included, mainly with a retrospective design, are the principal limitations of the present systematic review. First of all, despite the number of patients eventually enrolled seeming to be quite large, the criteria used to divide them led to the creation of small subgroups. As a consequence, the low number of cases per group may have not been sufficient to reach definitive conclusions regarding the best treatment options for TEF. It can not be excluded that dividing patients differently may have changed the conclusion.

In addition, comparison among groups in 12 articles is limited, as populations are not comparable in terms of demographic characteristics (age). Furthermore, the different follow-up lengths reported by different studies may influence the evaluation of clinical results. In addition, PROM results were reported by few studies, limiting the potential of comparison between different surgical techniques and drawing conclusions relative to the optimal surgical approach.

## 5. Conclusions

Better clinical results with low complication rates were achieved with ORIF surgery rather than ARIF; when analyzing arthroscopic procedures, suture fixation resulted in higher PROMs compared to screw fixation procedures and reduced incidence of postoperative complications.

However, there is insufficient evidence to define the superiority of open versus arthroscopic fixation or screw versus suture fixation procedures; therefore, the best treatment option should be based on the surgeon’s skills and personal preferences.

## Figures and Tables

**Figure 1 children-10-01379-f001:**
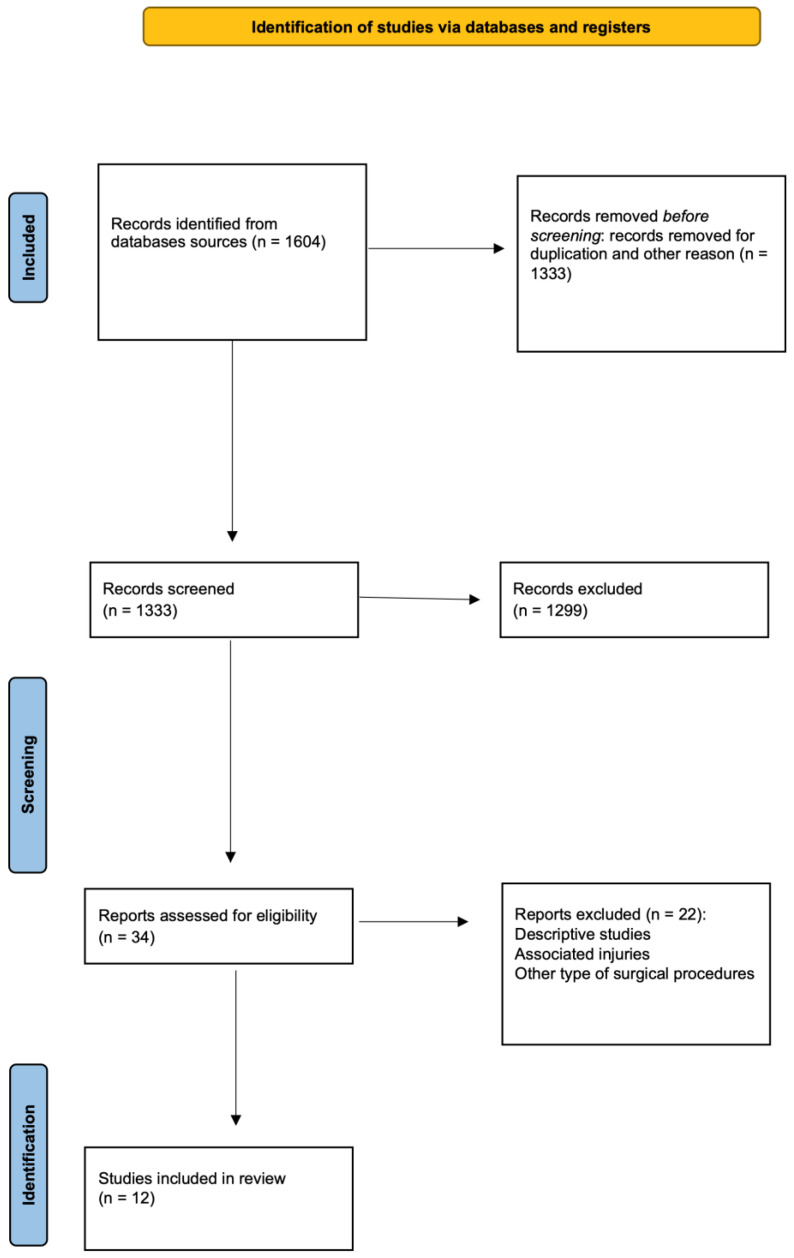
PRISMA diagram showing the systematic review process.

**Table 2 children-10-01379-t002:** MINORS scores of the included studies.

	A Clearly Stated Aim	Inclusion of Consecutive Patients	Prospective Collection of Data	Unbiased Assessment of the Study Endpoint	Endpoints Appropriate to the Study Outcomes	Follow-Up Appropriate	Loss to FU < 5%	Calculation of the Study Size	Adequate Control Group	Contemporary Groups	Baseline Equivalence of Groups	Adequate Statistical Analysis	Total
Zheng (2021) [11]	0	0	2	2	2	2	2	0	2	2	2	2	18
Edmonds (2015) [12]	2	0	0	2	2	2	2	0	2	2	2	2	18
Watts (2016) [13]	2	0	0	2	2	2	2	0	2	2	2	2	18
Xu (2016) [14]	2	0	0	2	2	2	2	0	NA	NA	NA	NA	10
Zhao (2018) [15]	2	2	2	2	2	2	2	0	2	2	2	2	22
Callanan (2019) [16]	2	2	2	2	2	2	2	0	2	2	2	2	22
Çağlar (2021) [17]	2	0	0	2	2	2	2	0	NA	NA	NA	NA	10
Russu (2021) [18]	2	0	2	2	2	0	2	0	NA	NA	NA	NA	10
Honeycutt (2020) [19]	2	0	2	2	2	2	1	0	NA	NA	NA	NA	11
Chalopin (2022) [20]	2	0	2	2	2	2	1	0	NA	NA	NA	NA	11
Zhang (2020) [21]	2	2	2	2	2	2	2	0	NA	NA	NA	NA	14
Quinlan (2021) [22]	2	2	2	2	2	2	1	0	NA	NA	NA	NA	13

**Table 3 children-10-01379-t003:** *p*-values were calculated using Student’s t-test (continuous variables) or chi square test (categorial variables) comparing ORIF and ARIF treatment. Data are reported as mean and standard deviation (SD) for continuous variables and as numbers for categorical variables. Bold is used to identify statistically significant results.

Variables	ORIF	ARIF	*p*-Value
Follow up	50.8 SD 31.6	44.7 SD 22.5	0.08
Age (years)	11.7 SD 0.6	12.2 SD 1.5	**0.0134**
Men	40	193	0.123
Sport injury	2	79	0.173
Days trauma	6.3 SD 2.9	8.6 SD 5.9	**0.0055**
Tegner score	7.8 SD 0.9	7.05 SD 0.9	**0.0196**
KT1000	NA	3.4 SD 2	-
IKDC Score	92.1 SD 3.6	89.7 SD 5.2	0.1550
Lysholm score	96.7 SD 3.2	91.4 SD 4.7	**<0.0001**
Arthrofibrosis	4	32	**0.034**
Implant removal	1	26	**0.006**

**Table 4 children-10-01379-t004:** *p*-values were calculated using Student’s t-test (continuous variables) or chi square test (categorial variables) comparing arthroscopic suture and arthroscopic screw treatment. Data are reported as mean and standard deviation (SD) for continuous variables and as numbers for categorical variables. Bold is used to identify statistically significant results.

Variables	Arthrosuture	Arthroscrew	*p*-Value
Follow up	47 SD 24.4	36.6 SD 17.2	**<0.0001**
Age (years)	12.4 SD 1.4	11.3 SD 1.1	**<0.0001**
Men	144	38	0.373
Sport injury	51	28	**0.0001**
Days trauma	9.24 SD 6.4	6.3 SD 0	**0.01**
Tegner score	7.5 SD 0.9	6.4 0.5	**<0.0001**
KT1000	3.4 SD 2	NA	-
IKDC Score	90.4 SD 5.6	86.7 SD 5.8	**0.0034**
Lysholm score	92.5 SD 4.6	86.2 SD 4.5	**<0.0001**
Arthrofibrosis	13	12	**0.0152**
Implant removal	3	22	**<0.00001**

## Data Availability

The data presented in this study are openly available upon request to the corresponding author.

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
