# Peer review of "Management and Outcomes of Tibial Eminence Fractures in the Pediatric Population: A Systematic Review"

_children, 2023, doi:10.3390/children10081379_

Round 1

Reviewer 1 Report

This is a good systematic review with metanalysis. 

The main issue concerns the abstract. If I read only the abstract, it is hard to understand why the conclusion follows such results. From the results you wrote, the logical conclusion should be : "ORIF should be preferred. When using arthroscopic procedures, suture fixation should be preferred". 

In my opinion, you should add some lines concerning the scientific "power" to the results section of the abstract to motivate your conclusion. You should also write some numbers and data instead of "better" or "more": mean values, SD, %, p-values, odds ratios....

Minor revisions:

In the discussion, you wrote  the details of various scores. This should go to methods section. 

Reviewer 2 Report

Thank you for submitting your systematic review on pediatric tibial eminence or spine fractures. You did a thorough review but did not reach definitive conclusions despite a large number of involved extremities. I feel you review has potential for publication. I only have a few questions to be addressed in a revision.

Title

(1). I have always felt that the appropriate anatomical term was tibial eminence (anterior and posterior) rather than spine. The former seems more descriptive. Please review current anatomy texts to determine which is the most appropriate. if you choose the former it will need to changed throughout your text.

Introduction

(2). Page 2, Line 50. Since your review deals with treatment I would suggest a statement that in Meyers-McKeever types 1 and type 2 the knee is typically immobilized in extension to reduce the fracture gap and enhance healing. This way you will have covered treatment for the fracture types.

Materials and Methods

(3). Page 4, Line 104. I was surprised that out of 1333 articles only 12 were suitable for your review. While I agree with your rationale and selection I would emphasize why this was performed. Although your number of patients included were relatively large (381 knees) it seemed small once they were divided into the various groups for comparison. This is probably why definitive conclusions could not be made? You may want to include this in the next to last paragraph of the Discussion.

Round 2

Reviewer 1 Report

Thank you for revising!